# Antiemetic Drugs Compatibility Evaluation with Paediatric Parenteral Nutrition Admixtures

**DOI:** 10.3390/pharmaceutics15082143

**Published:** 2023-08-15

**Authors:** Szymon Tomczak, Maciej Chmielewski, Jagoda Szkudlarek, Anna Jelińska

**Affiliations:** Department of Pharmaceutical Chemistry, Poznan University of Medical Sciences, 6 Grunwaldzka, 60-780 Poznań, Poland

**Keywords:** compatibility, ondansetron, supportive drugs, hydrocortisone, dexamethasone, parenteral nutrition, interaction

## Abstract

Chemotherapy-induced nausea and vomiting are defined as the most common of side effects of treatment and, at the same time, are very difficult to accept for patients’, frequently causing changes in the therapy regimen, significantly reducing its effectiveness. Thus, an antiemetic prophylactic is essential to the provision of such a therapy for the patient. Pharmacotherapy often includes various drugs, including antiemetics, with the administration of such drugs by injection through two separate catheters being the preferred method. However, the co-administration of drugs and parenteral nutrition admixtures (PNAs) requires the consideration of compatibility, stability and potential negative interactions. To meet the purposes of clinical pharmacy, a compatibility test of ondansetron, dexamethasone and hydrocortisone with paediatric PNAs was conducted. PNAs differ in the composition of amino acid source (Primene^®^ or Aminoplasmal Paed^®^ 10%) and the type of injectable lipid emulsion (Lipidem^®^ 200 mg/mL, Clinoleic^®^ 20%, SMOFlipid^®^ 200 mg/mL, Intralipid^®^ 20%). An in vitro evaluation was performed in a static way as a simulated co-administration through a Y-site. The drug PNA ratios were determined based on the extreme infusion rates contained in the characteristics of medicinal products. All calculations were performed for a hypothetical patient aged 7 years weighing 24 kg. As a result of this study, it can be concluded that all tested PNAs showed the required stability in the range of parameters such as pH, osmolality, turbidity, zeta potential, MDD and homogeneity. The co-administration of antiemetic drugs does not adversely affect lipid emulsion stability. This combination was consistently compatible during the evaluation period.

## 1. Introduction

Postoperative vomiting, often combined with postoperative nausea (PONV), is considered to be one of the most common side effects of general anaesthesia, affecting approximately 30% of randomly selected patients and as many as 70–80% of patients classified as high-risk [1,2]. An area where there is an even greater problem of vomiting and accompanying nausea is associated with anti-cancer treatment. Chemotherapy-induced nausea and vomiting (CINV) is defined as the most common of side-effects and at the same time most difficult to accept for patients’ in treatment, which causes changes in the therapy regimen, thereby significantly reducing its effectiveness [3,4]. Emetogenic agents, i.e., chemotherapy, induce nausea and vomiting by activating specific receptors and releasing neurotransmitters through peripheral or central mechanisms. These undesirable side effects can reduce patient’s quality of life and thus result in low adherence to treatment. Thus, correct antiemetic prophylaxis is essential in providing the patient with effective treatment with high compliance and an overall better quality of life. For vomiting and nausea prophylaxis drugs may be included 5-HT3 receptor antagonists, NK-1 receptor antagonists, D2 receptor antagonists, and glucocorticosteroids [4,5].

Pharmacotherapy, including antiemetic treatment, is very often a complex procedure that requires the utilisation of various drugs. As a consequence of the complications that can occur, therapy regimens are growing in their complexity One-third of patients have been prescribed three injectable drugs to be administered at the same time [6], which in turn has limited parenteral access. In the absence of confirmed compatibility in the pharmaceutical phase, these drugs must be administered through separate catheters, however due to the dosing regimen and frequent administration by continuous infusion, this practice becomes not only impractical but also impossible to administer. These limitations mainly affect neonatal and paediatric intensive care units (NICU and PICU, respectively), where patients can only tolerate the insertion of a single, double or, as a maximum, triple lumen of central venous catheters. Co-administration using a Y-site or the direct injection of drugs after drugs, poses a significant risk of negative interaction. The problem is that the delivered solutions that come into contact with each other in the lumen may not be compatible. This may manifest as crystallisation, catheter blockage, discoloration, turbidity, gas formation, inactivation, drug breakdown and or lipid emulsion breakdown. Unfortunately, this practice poses a serious threat to the health and life of the patient [7,8].

Anticancer therapy, and cancer itself, reduce the patient’s vitality. They are often malnourished by treatment-related adverse reactions, such as lack of appetite, changed flavour perception, premature sense of satiety, mucosal ulcers, nausea, and vomiting [9,10,11,12]. In such patients, it is advisable to include clinical nutrition depending on the state of health and other individual patient needs. In the case of the supply of nutrients by the enteral route being insufficient or even impossible, parenteral nutrition should be included [9,13]. An important advantage of parenteral nutrition is the possibility of personalising the composition, ensuring all the necessary macro and microelemental needs are met. PNAs, due to their biphasic composition (oil-in-water emulsion), are susceptible to interactions. Lipid emulsions are non-transparent, and some kinds of interaction were difficult to notice at an early stage of evaluation. The risk of the combined supply of drugs and PNA is therefore greater, and the lack of sufficient scientific data on this important issue necessitates the provision of new data.

To meet the purposes of clinical pharmacy, we decided to test the compatibility of ondansetron, dexamethasone and hydrocortisone with paediatric PNAs. Studied drugs were used in the treatment and prevention of PONV and CINV, whereas PNAs differ in the composition of amino acid source (Primene^®^ or Aminoplasmal Paed^®^ 10%) and the type of injectable lipid emulsion (Lipidem^®^ 200 mg/mL, Clinoleic^®^ 20%, SMOFlipid^®^ 200 mg/mL, Intralipid^®^ 20%). An in vitro evaluation was performed, simulating co-administration through a Y-site. To the best of our knowledge, no study has extensively explored the potential interactions of such a combination.

## 2. Materials and Methods

### 2.1. Parenteral Nutrition Admixtures and Drug Solutions

Antiemetic drugs were used:Hydrocortisone (HC)—Corhydron 100 mg (hydrocortisone sodium succinate), powder and solvent for solution for injection and infusion (Bausch Health Ireland Limited, Dublin, Ireland);Dexamethasone (DEX)—Dexaven 4 mg/mL (dexamethasone phosphate), solution for injection (Bausch Health Ireland Limited, Dublin, Ireland);Ondansetron (OND)—Ondansetron Accord 2 mg/mL (ondansetron hydrochloride dihydrate), solution for injection and infusion (Accord Healthcare, Middlesex, UK).

According to the manufacturer’s advice, all drugs were reconstituted in 0.9% normal saline solution (Polpharma, Starogard Gdański, Poland) to reach the final concentration of 0.98 mg/mL (HC), 0.08 mg/mL (DEX) and 0.02 mg/mL (OND).

Eight parenteral nutrition admixtures based on two different amino acid preparations (Aminoplasmal Paed 10% and Primene^®^ 10%) and four types of lipid emulsion (Lipidem^®^ 200 mg/mL, ClinOleic^®^ 20%, SMOFlipid^®^ 200 mg/mL and Intralipid^®^ 20%) were analysed in the study. The compositions of the admixtures are presented in Table 1. The names of PNA refer to the first letter of ILE (L, C, S and I) and amino acid sources (A and P). To simulate administration through the Y-site connector, the rate was determined for each of the tested drugs and the PNAs based on the extreme rates of administration contained in the characteristics of medicinal products. All calculations were performed for a hypothetical patient aged 7 years weighing 24 kg. As a result, the drug/PNA volume ratios (V/V) were determined, which are as follows: HC 1:1 and 4:1, DEX 1:1 and 2:1, and OND 1:1 and 2:1.

### 2.2. Compatibility Testing

Drug solution and PNAs were mixed based on the calculation mentioned above. The obtained samples were tested on two end points: immediately after preparation (0 h) and after 4 h. During the test, the samples were stored at 20 °C ± 2 °C with light access.

The following measurements were performed:Visual control;pH;Osmolality;Particle size;Zeta potential;Turbidity for a lipid-free admixture.

Each measurement was performed in triplicate (*n* = 3) and expressed as a mean with a standard deviation.

#### 2.2.1. Visual Control

Each of the 10 mL of the samples stored in a plastic test tube was assessed visually without the aid of equipment by two independent researchers according to Ph. Eur.’s recommendation [14] for:Color change;Delamination;Sedimentation;Gas formation;Aging processes.

#### 2.2.2. pH Measurement

The measurement was performed using a Mettler Toledo Seven Compact pH/Ion S 220 pH meter (Mettler Toledo, Columbus, OH, USA). Before starting the measurements, the instrument was calibrated. The electrode of the pH meter was placed directly in the plastic tube with the investigated sample. Between measurements, the electrode was rinsed with distilled water.

#### 2.2.3. Osmolality Measurement

100 µL of the sample was transferred to Osmo-Krio tubes. Measurements were made on an osmometer 800CLG (TridentMed, Warsaw, Poland), and the principle of operation was based on the measurement of the freezing point of the analysed sample. The instrument was calibrated before starting the series of measurements using Osmometer Calibration Solution 800 cl, 0 mOsm/kg H_2_O, Cat. Yes. 800.02 (TridentMed, Warsaw, Poland). One osmol corresponds to 1 mole of a chemical compound exhibiting osmotic activity while dissolved in 1 kg of water.

#### 2.2.4. Measurement of Turbidity

The sample for turbidity measurement must be transparent, thus the composition from Table 1 was modified. The equal part of lipid injectable emulsion (ILE) was replaced with water for injection. Lipid-free PNA was only supplemented with trace elements. The vitamins, due to colour, had to be omitted. Ratios of drugs: PNA remained the same. Turbidity was measured using a TU52000 Laboratory Laser Turbidimeter (Hach Company, Loveland, CO, USA). A total of 10 mL of sample (drug-lipid-free PNA) in glass cuvettes was placed in a turbidimeter cell; next, measurements were performed in triplicate. The results are expressed as the mean value with standard deviation.

#### 2.2.5. Measurement of Particle Size, PDI and Zeta Potential

All three parameters were analysed using the ZetaSizer Nano ZS apparatus (Malvern Instruments Ltd., Malvern, UK) in the same run-time. The solutions were prepared by mixing 1 mL of sample with 9 mL of distilled water and were transferred using sterile syringes to the DTS1070 cuvette. Measuring the emulsion particle size enables the determination of the MDD, mean particle size (based on their diameter in nm), and polydispersity index (PDI), expressing the degree of heterogeneity of the particles. Additionally, the electrokinetic potential between emulsion phases was expressed as zeta potential in mV.

## 3. Results

Compatibility studies were carried out for three antiemetics drugs with eight PNAs. The composition was adapted to a paediatric patient aged 7 years weighing 24 kg. The drugs were dissolved in a normal saline solution and mixed with PNAs in volume ratios simulating the worst case scenario during Y-site administration.

The source of amino acid influenced the pH of the tested admixtures. The Primene-based admixtures ranged from 5.50 to 5.61, while those containing Aminoplasmal Paed had higher pH values of 6.11 to 6.32 (Table 2).

The addition of OND does not influence this parameter, but when comparing pH values immediately after mixing, a slight drop was observed for most samples containing steroids. The addition of dexamethasone caused a fractional increase in the pH of the Primene-based PNAs, reaching pH = 6.25 (DEX:IP, 2:1, 0 h). The addition of hydrocortisone broadens the pH range of the PNAs, regardless of the amino acid source. The increase was up to 6.25 for HC Primene 0 h. Despite this increase, after mixing the drugs with PNAs, no significant changes (*p* > 0.05) in pH values during the evaluation period were observed (Figure 1).

The osmolality of PNAs without drugs ranged from 1318 to 1485 mOsm for Aminoplasmal-based PNA and with a wider range for Primene-based ones (1401–1482 mOsm). A concentration-dependent decrease in osmolality was observed due to the dissolved sample with the low-osmolality drug solution. After four hours of storage, the observed osmolality changes for the drug/PNA samples did not exceed ±2% (Figure 2). The turbidity for all analysed samples remained low, below 0.19 NTU. No differences in lipid PNA or drug PNA were observed during the evaluation period.

The zeta potential values for blank samples varied from −11.1 mV to −16.7 mV and correlated with the lipid source (Figure 3). The highest values were observed for PNAs with and without drugs containing Clinoleic. The drug addition to all PNAs resulted in a significant decrease in zeta potential (*p* < 0.05). The biggest difference was captured for the HC sample mixed with SP in the ratio of 4:1 at 0 h (15.0 mV difference). However, despite such a large difference, these values remained stable throughout the experiment (4 h), with a change below the acceptance criterion (Δ = 3.7 mV < 5.0 mV).

Similarly to the zeta potential, the size of the particles strictly depends on the lipid source. The smallest values were for the Lipidem-based PNAs (Range: 230–246 nm), and the highest (324–338 nm) were for the SMOFlipid^®^ based PNAs. The addition of drugs did not significantly change lipid droplet size. The MDD varied slightly after 4 h of storage and was well below the recommended cut-off of 500 nm for injectable emulsions [15] (Figure 4).

No second fraction of lipid droplets >1000 nm was observed for any of the samples (Figure 5). The PDI values did not correlate with the composition, drug or time passed after addition. The values for all samples were below 0.19. This is indicative of a narrow size distribution, which is desirable in connection with emulsion stability.

## 4. Discussion

The administration of drugs by injection through two separate catheters is always the preferred method. However, the co-administration of drugs and PNAs requires the consideration of compatibility, stability and potential negative interactions. The assessment of the literature data is insufficient to be conclusive. Therefore, it is important to examine the effect of the co-administration of drugs and PNAs in the context of physicochemical changes. Important elements are the type of lipid emulsion, the concentration of polyvalent ions, drug concentration, the type of liquid for reconstitution and the drug admixture ratio that best reflects clinical realities.

So far, the compatibility data of antiemetics and PNA are scarce and often conflicting. Ondansetron has been proven to destabilise lipid emulsion with oiling out [16], but in contrast, OND’s compatibility has been confirmed by other researchers [17,18,19]. Staven et al. [20] confirmed the compatibility of ondansetron and dexamethasone with ready-to-use PNAs, namely Numeta G16E and Olimel N5E. The result of the analysis of data available at the hospital in Sao Paulo, Brazil, provides information on the compatibility of dexamethasone with PNA and lipid-free parenteral nutrition in the concentration range of 1–4 mg/mL [21]. Hydrocortisone data confirm its compatibility with lipid-free PNs [22,23,24] and total parenteral nutrition [16]. Such a lack of concrete knowledge indicates the importance of expanding the research and filling the knowledge gap, especially with different types of ILE.

In our research, we examined three antiemetic drugs used as supportive drugs in anticancer treatment. To simulate clinical conditions, the ratios in which the drug and PNA would be in contact in the lumen were calculated. On this basis, two extreme ratios were selected for further research in order to best capture potential interactions. The study was performed in two-time endpoints: 0 and 4 h. However, it should be remembered that the contact between the drug and the admixture in the common catheter is much shorter and may be counted in minutes.

Due to the lack of guidelines for summarising the compatibility tests procedure, it was decided to establish the following acceptance criteria: no changes visible to the unaided eye during the entire test period, while during the evaluation period (4 h), pH changes no greater than 0.2, osmolality changes less than 5%, zeta potential differences lower than 5 mV, MDD below 500 nm and the absence of second fraction of lipid droplets higher than 1000 nm, and turbidity differences lower than 0.5 NTU [19,25,26,27,28].

Osmolality refers to the content of osmotic active compounds in a sample, and its changes may indicate incompatibilities. This is indicated by the precipitation of some ions from the solution. Osmolality is also crucial for the route of parenteral administration. Preparations below 1000 mOsm can be administered to peripheral vessels, while those with higher values must be administered to central vessels. Therefore, the observance of such a large decrease in osmolality for the tested samples, in which the ratio of the drug solution in normal saline is significant. None of the analysed samples exceeded the acceptance criterion of 5% during the evaluation period. Another parameter used in the compatibility assessment was the pH, which can show changes in the lipid emulsion, such as the release of free fatty acids from triglyceride hydrolysis. Mixing a liquid with a significantly different pH may affect the lipid emulsion and cause destabilisation. The solubility of drugs is also pH-dependent. An overly large difference may lead to precipitation or oiling out. Due to the presence of electrolytes and amino acids, PNA has a certain buffer capacity, which can neutralise the negative impact of external solutions. The influence of the amino acids constituting PNAs on pH is well observable in the tested admixtures despite the equal content of the sum of amino acids (10 g/100 mL) and the similar content of nitrogen per 100 mL (1.52 g and 1.5 for Priemene and Aminoplasmal Paed, respectively). These amino acid sources differ in the content and composition of individual amino acids, with Primene containing more acidic and less alkaline amino acids than Aminoplasmal Pead. Thus, Primene-based emulsions are more acidic and closer to value when the lipid emulsion below 5.5 is prone to destabilisation. The addition of drugs to the tested mixtures increased the pH, except in the case of ondansetron. The pH of the drug solution was the lowest (4.37 ± 0.01) of all the tested drugs. It is soluble in water but its solubility decreases when the pH is >5.7 [29]. This may be important for Aminoplasmal Paed-based PNAs with a pH ranging from 6.11 to 6.32. However, no changes in pH during the evaluation were observed, and also the turbidity measurement did not confirm drug precipitate for any sample. Changes greater than 0.5 NTU were not observed, while the maximum turbidity value for the tested drugs was 0.197 NTU for DEX with Primene at the ratio of 4:1.

The parameter determining the stability of the emulsion system is the zeta potential. It describes the difference in electrokinetic potential between the two phases of the emulsion. The content of the electrolytes and the type of emulsifier used affect the value of this parameter. The zeta potential of lipid emulsions, based on phospholipids as emulsifiers, ranged from −50 mV to −40 mV [30]. Amino acids and mono- (potassium and sodium) and bivalent (magnesium and calcium) ions addition leads to a decrease [31,32]. The greater the difference, the greater the absolute value, and the greater the interaction forces, which is equivalent to the stability of the emulsion as a suitable dispersion of lipid droplets. The obtained results were in the range from −30.3 to −12.3 mV, which was classified as stable and moderately stable [30]. The addition of drugs caused a decrease in the value of the zeta potential, moving away from zero, thus increasing the stability of the system. The literature data show that adding drugs to PNAs influenced changes in the zeta potential. Differences depend on the added drugs: Ondansetron [18] caused a decrease in absolute value zeta potential. Contrarily, adding sodium valproate [33] to the PNAs caused its increase.

The crucial parameter in safety therapy is the size of the lipid droplets. As mentioned above, the lipid emulsion can interact with particle aggregation. The administration of large micelles or their groupings can have dangerous consequences and affect the health and even the life of the patient. A known complication is the embolism of small vessels, e.g., of the retina, liver or brain. According to the United States Pharmacopeia (USP), lipid droplet sizes expressed as intensity-weighted MDD must not exceed the pharmacopoeial limit of 500 nm [15]. The MDD of the analysed samples depended on the injectable lipid emulsion source. The addition of drugs and time of storage did not increase this parameter, and all samples are in agreement with USP limits. The second fraction of lipid droplets was not observed (Figure 2). The results are backed by a polydispersity coefficient, which was below 0.19. This being indicative of narrow distribution, which is desirable considering emulsion stability.

## 5. Conclusions

As a result of the tests of the three antiemetic drugs with 8 paediatric PNAs, it can be concluded that all tested PNAs showed the required stability in the range of parameters such as pH, osmolality, turbidity, zeta potential, MDD and homogeneity. The co-administration of antiemetic drugs (HC, DEX and OND) does not adversely affect lipid emulsion stability. This combination was compatible during the evaluation period. However, it must be taken into account that a change in the composition of the admixture or excipients may affect the stability, hence more study in this area is required to be able to unequivocally implement these results into clinical practice.

## Figures and Tables

**Figure 1 pharmaceutics-15-02143-f001:**
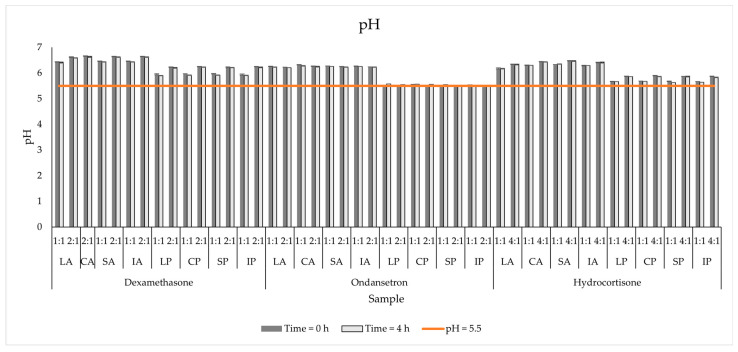
The pH measurements results for mixed PNAs with drug solutions.

**Figure 2 pharmaceutics-15-02143-f002:**
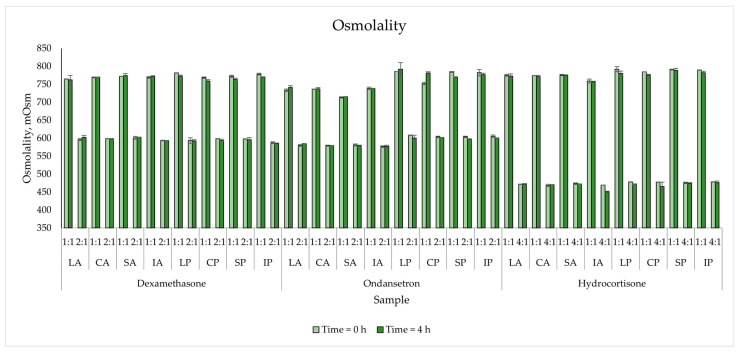
The osmolality measurements results for mixed PNAs with drug solutions.

**Figure 3 pharmaceutics-15-02143-f003:**
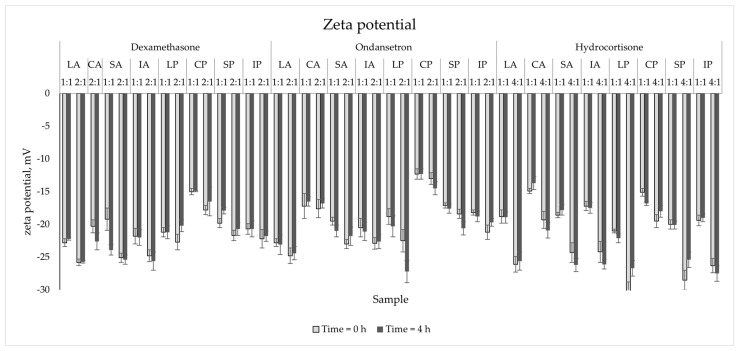
The zeta potential measurements results for mixed PNAs with drug solutions.

**Figure 4 pharmaceutics-15-02143-f004:**
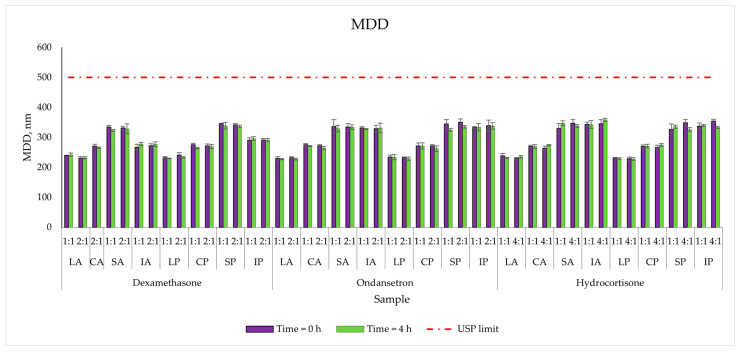
The lipid droplet size measurements results for mixed PNAs with drug solutions.

**Figure 5 pharmaceutics-15-02143-f005:**
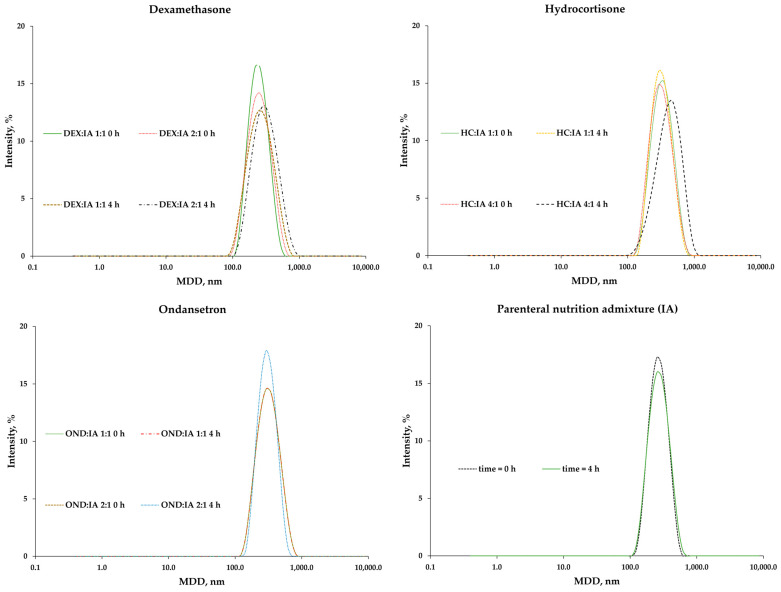
Lipid droplet size intensity for Intralipid^®^ 20%-Aminoplasmal Paed^®^ 10% IA PNA with and without drugs.

**Table 1 pharmaceutics-15-02143-t001:** The composition of parenteral nutrition admixtures.

Components	Drugs	Manufacturer	Parenteral Nutrition Admixture							
			LA	CA	SA	IA	LP	CP	SP	IP
Carbohydrates	Glucose 40%	B. Braun, Melsungen, Germany	432.0							
Amino acids	Aminoplasmal Paed^®^ 10%	B. Braun, Melsungen, Germany	240.0				-			
	Primene^®^ 10%	Baxter, Warsaw, Poland	-				240.0			
ILE	Lipidem^®^ 200 mg/mL	B. Braun, Melsungen, Germany	240.0	-	-	-	240.0	-	-	-
	Clinoleic 20%	Baxter, Warsaw, Poland	-	240.0	-	-	-	240.0	-	-
	SMOFlipid^®^ 200 mg/mL	Frasenius Kabi AB, Uppsala, Sweden	-	-	240.0	-	-	-	240.0	-
	Intralipid^®^ 20%	Frasenius Kabi AB, Uppsala, Sweden	-	-	-	240.0	-	-	-	240.0
Water	Aqua pro injectione	Polpharma, Starogard Gdański, Poland	240.0							
Electolytes	Inj. Natrii Chlorati 10%	Polpharma, Starogard Gdański, Poland	14.1							
	Kalium Chloratum 15%	Polpharma, Starogard Gdański, Poland	12.0							
	Inj. Magnesii Sulfurici 20%	Polpharma, Starogard Gdański, Poland	3.0							
	Calcio gluconatio 1000 mg/10 mL	Galenica Sense, Siena, Italy	21.3							
	Glycophos^®^ 216 mg/mL	Frasenius Kabi AB, Uppsala, Sweden	4.8							
Trace elements	Peditrace^®^	Frasenius Kabi AB, Uppsala, Sweden	1.2							
Vitamins	Cernevit^®^	Baxter, Warsaw, Poland	0.6							
Total volume			1209.0							

ILE—Lipid injectable emulsion; L/C/S/IA—Aminoplasmal-based PNAs; L/C/S/IP—Primene-based PNAs.

**Table 2 pharmaceutics-15-02143-t002:** Drug solutions and PNA without drugs—summary of parameters.

		pH	Osmolality, mOsm	Turbidity, NTU	PDI	ZP, mV	MDD, nm
Drug	DEX	7.48 ± 0.01	284.5 ± 0.7	0.153 ± 0.008	N/A	N/A	N/A
	HC	7.41 ± 0.01	294.3 ± 1.5	0.164 ± 0.008	N/A	N/A	N/A
	OND	4.37 ± 0.01	284.7 ± 0.6	0.184 ± 0.014	N/A	N/A	N/A
PNA	LA	6.21 ± 0.01	1419.5 ± 7.8	0.164 ± 0.008	0.070 ± 0.011	−16.3 ± 0.8	237.5 ± 5.3
	CA	6.24 ± 0.01	1429.0 ± 0.0	0.165 ± 0.026	0.089 ± 0.009	−12.2 ± 0.1	279.8 ± 3.3
	SA	6.21 ± 0.01	1442.0 ± 7.1	0.132 ± 0.021	0.096 ± 0.027	−15.1 ± 0.1	331.1 ± 6.9
	IA	6.21 ± 0.01	1443.5 ± 0.7	0.132 ± 0.016	0.093 ± 0.023	−16.2 ± 0.6	278.9 ± 2.7
	LP	5.54 ± 0.01	1444.0 ± 19.8	0.178 ± 0.014	0.059 ± 0.025	−16.6 ± 0.7	236.4 ± 4.1
	CP	5.56 ± 0.01	1412.5 ± 13.4	0.173 ± 0.018	0.090 ± 0.014	−11.6 ± 0.7	277.6 ± 5.3
	SP	5.55 ± 0.01	1446.0 ± 21.2	0.112 ± 0.012	0.093 ± 0.021	−13.5 ± 0.3	331.7 ± 2.6
	IP	5.55 ± 0.01	1456.0 ± 0.0	0.134 ± 0.023	0.103 ± 0.026	−15.7 ± 0.5	301.7 ± 12.8

NTU—Nephelometric turbidity unit; N/A—not applicable; PDI—polydispersity index; ZP—zeta potential; MDD—mean droplet diameter; DEX—dexamethasone; HC—hydrocortisone; OND—ondansetron; PNA—parenteral nutrition admixtures; L/C/S/IA—Aminoplasmal-based PNAs; L/C/S/IP—Primene-based PNAs.

## Data Availability

The data presented in this study are available upon request from the authors.

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
