# Peer review of "Antiemetic Drugs Compatibility Evaluation with Paediatric Parenteral Nutrition Admixtures"

_pharmaceutics, 2023, doi:10.3390/pharmaceutics15082143_

Round 1
Reviewer 1 Report
The compatibility data of antiemetics and parenteral nutrition admixures are lacking, and the topic could be interested to the readers.
Abstract should be more concised. Some sentences could be replaced by common terms, e.g. prophylactic treatment with the correct antiemetic drug - correct antiemetic prophylaxis; pharmacotherapy, including antiemetic treatment, is very often a complex procedure utilizing various drugs - pharmacotherapy often includes various drugs including antiemetics. The statement in the first sentence that CINV causes changes should be replaced by ...can cause changes. Do not use abbreviation before mentioned the full term (e.g. PONV, PNA, HCT, DEX, OND).
Introduction should be focused on the topic - to omit postoperative vomiting; line 50 - the drugs are not used in the treatment but in the prophylaxis as well; line 54 - not only in hospital wards but can be in outpatient clinics. Malnoutrition is much more complexed in cancer patient than treatment-related (line 67-68). Again, abbrevaitions are used wrongly.
Materials and methods - Please do not separate in line measurements (2.2.) and visual control components (2.2.1.). What is the reason to choose a 7-yr-old child and a 4-hour period? Is additional time required?
Results - Do not repeat methods.
Discussion - Instead repeating sentences more attention should be placed to the comparison of the results.
? coadministration or co-administration, y- or Y-site, glycocorytosteroids, too-large etc.
Author Response
Dear Reviewers
Thank you for these quality comments. The manuscript has been carefully revised according to your suggestions. Al the changes made have been highlighted in yellow throughout the manuscript. Below please find our responses and changes that have been made.
- We reorganized the abstract to be more consistent.
- All abbreviations were corrected and now are DEX – dexamethasone, OND – ondansetron and HC – hydrocortisone.
- Mistakes and errors were corrected and the whole manuscript was read by native speakers.
- We decided to choose 7 years old kids only for methodology reasons. The parenteral nutrition admixture composition was prepared to meet all needs of these pediatric patients. The composition was based on our cooperation with a hospital pharmacy from the local university hospital.
- The second endpoint in our methodology (t2=4 h) is of course longer than the real contact time of fluids in the common line of the catheter. In fact, it lasts up to a few minutes, but this endpoint was stated based on the general methodology in this area, performed by other researchers, and allows captured potential interaction which cannot occur immediately.
- Results paragraph. We cut some methodology sentences and changed it to focus more on obtained data.
- To improve readability we extracted 4 figures from one (previous Figure 1), and changed the bars, descriptions, and colors for better understanding.
- Table 1. We explained used abbreviations and modified some names. All names used in column Drugs are brand names and we do not translate to English for example Aqua pro injection instead of Water for injection.
- Table 2. We added to the table legend all abbreviations used in the main body.
- Point 2.2. Compatibility testing and next ones 2.2.1 etc. We decided to leave them pointed separately for each measurement for better understanding and readability.
- Figure 5 (previous 2) We modified it using different linetypes.
We are grateful for your constructive criticism, and we hope that the changes introduced in the version that we are now submitting may be considered enough to improve the quality of our paper.
Best regards
Reviewer 2 Report
I have no major comments of the methodology that was used in this study. Only the examination of the parameters at 4 hours does seem to be clinically less relevant as mentioned in the text (drugs will be co-administered in a Y-catheter and only be in contact in a tube for a very short time).
Could the authors please add detail on the exact way the drugs were mixed (what type of container, what composition, brand...), unless I missed it, I cannot find those details.
unarmed eye > unaided eye?
in triplicated > in triplicate
table 2: the 'O' in 'Osmolality' has partially diappeared
some minor grammatical errors in some parts of the text
Author Response

(The authors gave the same response as above.)

Reviewer 3 Report
The authors address an important issue. The treatment of children with malignant diseases is complex and their vascular access is often limited to a single broviac catheter (with 1-3 lumens depending on their disease and age). As the parenteral nutrition has often to be stopped in order to allow infusion of antimicrobial (or even worse antifungal) medicaments or transfusions, studies examining parallel compatibility are direly needed in order to improve patient care.
This reviewer has several points for the authors to consider to their otherwise well-conducted work.
Abstract: Please avoid any abbreviations that were not introduced and explained in the abstract, e.g. PNA, PONV and many more.
Line 52: What is glycocorytosteroids?
Line 90: HCT is a common abbreviation for hydrochlorothiazide, assigning it to hydrocortisone results in relevant ambiguity and might confuse readers. Please avoid it.
Line 93: The standard abbreviation for dexamethasone is DEX. It is unclear to this reviewer why the authors prefer making up their own ones (which they don't consistently follow afterwards).
Table 1: Please user proper English in the description of the drugs if it is not a brand name used. Besides, some brandes are marked as such, but others (e.g. Lipidem) are not. Please be consistent, too.
Lines 136/141-142/144-145/159: It remains unclear why manufacturers of these products are not provided, why it has been done for almost all other materials.
Line 169: Why has the mixture to be checked to be homogenous and homogenous? Wouldn't it be sufficient if it were homogenous?
Table 2: Please repeat the lots of abbreviations in the table lengend to ensure that it is able to stand on its own.
Figure 1: Please consider separating it into different figures to improve readability. Moreover, the color choice for osmolality is not suitable as it would be hard for readers having tritanopia to be able to differentiate the bars. Moreover, it would enhance readability of the figures if there were some gaps between the bars of 0 and 4 hours too.
Line 199: How can a drug be addicted to a mixture?
Figure 2: Please consider using different linetypes instead of different (and related) colors to enhance readability of the figures.
Line 209: According to the methods, correlation analyses have not been conducted. Please clarify.
Lines 225/227/231/267: Either use the abbreviation exclusively throughout or avoid it at all, but not switching between them. These inconsistencies need to be avoided.
Line 273/306: Now it is DEX and not DXM anymore?
Lines 293/297: The abbreviation USP has not been introduced.
Lines 302-303: General statements should not be part of the conclusion.
References: The self-citation rate is 17% and clearly increased by citations that just mention a fact reported in one of their previous reports (e.g. in lines 286-287), but are not relevant to the line of argument. This reviewer does not mind self-citations clearly necessary to clarify methods or which are substantial to the line of argument, even if they exceed the typical limit of 10% (DOI: 10.1007/s11192-020-03417-5), but in the present case, these necessities are not met.
Some editing especially for sentence composition and word choices would be beneficial.
Author Response

(The authors gave the same response as above.)

Round 2
Reviewer 1 Report
The authors responded to all major suggestions /criticisms.
OK